# Unexpected Diversity of *Wolbachia* Associated with *Bactrocera dorsalis* (Diptera: Tephritidae) in Africa

**DOI:** 10.3390/insects10060155

**Published:** 2019-05-31

**Authors:** Joseph Gichuhi, Fathiya M. Khamis, Johnnie Van den Berg, Sunday Ekesi, Jeremy K. Herren

**Affiliations:** 1International Centre of Insect Physiology and Ecology (icipe), Kasarani, Nairobi 00100, Kenya; jgichuhi@icipe.org (J.G.); fkhamis@icipe.org (F.M.K.); sekesi@icipe.org (S.E.); 2Unit for Environmental Sciences and Management, North-West University, Potchefstroom 2520, South Africa; johnnie.vandenberg@nwu.ac.za; 3MRC-University of Glasgow Centre for Virus Research, Henry Wellcome Building, Glasgow G61 1QH, UK

**Keywords:** *Wolbachia*, *Bactrocera dorsalis*, oriental fruit fly, *Wolbachia* surface protein, mitochondrial COI haplotype

## Abstract

*Bactrocera dorsalis* (Hendel) is an important pest of fruit-bearing plants in many countries worldwide. In Africa, this pest has spread rapidly and has become widely established since the first invasion report in 2003. *Wolbachia* is a vertically transmitted endosymbiont that can significantly influence aspects of the biology and, in particular, the reproduction of its host. In this study, we screened *B. dorsalis* specimens collected from several locations in Africa between 2005 and 2017 for *Wolbachia* using a PCR-based assay to target the *Wolbachia* surface protein *wsp*. Of the 357 individuals tested, 10 were positive for *Wolbachia* using the *wsp* assay. We identified four strains of *Wolbachia* infecting two *B. dorsalis* mitochondrial haplotypes. We found no strict association between the infecting strain and host haplotype, with one strain being present in two different host haplotypes. All the detected strains belonged to Super Group B *Wolbachia* and did not match any strains reported previously in *B. dorsalis* in Asia. These findings indicate that diverse *Wolbachia* infections are present in invasive populations of *B. dorsalis*.

## 1. Introduction

*Bactrocera dorsalis* (Hendel) (Diptera: Tephritidae) is amongst the most serious pests of cultivated fruits across Asia and Africa owing to its high adaptation, polyphagy, fecundity and the extent to which it causes yield and revenue losses [1]. Many other regions in the world are at risk of invasion and establishment of *B. dorsalis* [2,3]. Notably, *B. dorsalis* has been intercepted on more than 50 occasions since the 1980s in Florida and California, indicating that North America is at constant risk for the establishment of this pest [1,4]. In addition, there has been a recent report of *B. dorsalis* invading Europe, where is has been found in Italy [5]. In Africa, *B. dorsalis* was first detected in 2003 in Kenya and since then the pest has rapidly spread and established in most African countries often displacing the native *Ceratitis cosyra* (Walker) (Diptera: Tephritidae) as the primary fruit fly pest of mango [6,7]. The invasion of Africa by this pest had major consequences for fruit production, causing major losses in yield [8] as well as revenues [9].

*Wolbachia* is an intracellular bacterial parasite known to infect many arthropod species [10,11,12]. *Wolbachia* are maternally-transmitted in the egg cytoplasm, and therefore, have evolved a number of reproductive manipulations to increase the fitness of *Wolbachia*-infected matrilines. In many cases, *Wolbachia* cause cytoplasmic incompatibility between uninfected females and infected males. This ability to cause cytoplasmic incompatibility can result in *Wolbachia*-infected lineages rapidly increasing in frequency in a host population. The release of *Wolbachia*-infected incompatible males is potentially a very effective mechanism for decreasing pest insect populations (incompatible insect technique, IIT) [13]. IIT may have a number of benefits relative to the sterile insect technique (SIT) because radiation is not required. Notably, a symbiont-based pest management technique utilizing a *Wolbachia* strain that causes cytoplasmic incompatibility in fruit flies has been demonstrated in the Mediterranean fruit fly, *Ceratitis capitata* (Wiedemann) (Diptera: Tephritidae) [14] and evaluated for the olive fruit fly *Bactrocera oleae* (Rossi) (Diptera: Tephritidae) [15]. In addition, *Wolbachia-*induced cytoplasmic incompatibility can be used to spread symbionts and transgenes through target insect populations, which could be useful for controlling pests and blocking the capacity of vectors to transmit diseases [16,17,18,19]. Some *Wolbachia* strains have also been found to modify their host’s susceptibility to parasitoids [20]; therefore, knowledge of *Wolbachia* infection status can be of relevance to optimizing integrated pest control strategies employing parasitoid wasps.

*Wolbachia* are a diverse bacterial clade that has been broadly categorized into several super groups. Currently, there are at least 16 recognized super groups, designated A–F and H–Q [21,22,23,24,25,26,27,28,29,30]. Some strains show strong associations with certain host species, while others infect more than one host species and observations of multiple infections of same species or even same individuals are commonly reported [31,32,33,34,35,36,37]. This pattern indicates that over evolutionary timescales, horizontal transmission of *Wolbachia* is commonplace [38]. At the population level, *Wolbachia* are transmitted vertically and since mitochondria are co-inherited, this can establish a linkage disequilibrium between *Wolbachia* and the host mitochondrial haplotype [39].

In the Tephritidae family, several studies have detected *Wolbachia* strains in the genera *Rhagoletis* [32,40,41], *Anastrepha* [35,42,43,44,45,46], *Ceratitis* [13,47,48] *Dacus* [49,50,51] and *Bactrocera* [49,50,51,52,53,54]. In *B. dorsalis*, *Wolbachia* has been reported at low prevalence in populations from China [53] and Thailand [52,54]. The objective of the current study was to investigate the presence and diversity of *Wolbachia* strains in *B. dorsalis* populations in Africa and to evaluate the infection patterns of *Wolbachia* and associations with mtDNA haplotypes in populations of this host sampled between 2005 and 2017.

## 2. Materials and Methods

*Bactrocera dorsalis* male flies were collected using attract-and-kill bait stations with methyl eugenol as attractant and malathion as killing agent, placed in Lynfield traps. Trapping was done in mango farms in 2017 from Mwanza (S 2°43′01.3″ E 33°01′20.4″) and Morogoro (S 06°57′38.5″ E 037°31′59.1″) in Tanzania, Bunamwaya (N 0°16′17.8752″ E 32°33′25.6284″) in Uganda and Kassala (N 15°28′39.1728″, E 36°21′57.9204″), Gezira (N 14°36′29.4″ E 33°47′27.5″) and Singa (N 12°47′46.6″ E 33°11′51.5″) in Sudan. *Bactrocera dorsalis* female flies were retrieved from infested mango collected from mango farms in 2017 from Nguruman (S 01°48′32″ E 36°03′35″), Kitui (S 01°21′ E 38°00′), Muranga (S 0°42′50.0″ E 37°07′03.4″) and Embu (S 0°28′56.6″ E 37°34′55.5″) in Kenya. Infested fruit were dissected for third stage larvae to emerge and pupate in fine sterile sand. Puparia were sieved from the sand and maintained in ventilated perspex cages until eclosion. All samples were stored in absolute ethanol at −20 °C. DNA was extracted from each individual using the ISOLATE II Genomic DNA Kit (Bioline, London, UK). Voucher specimens collected between 2005 and 2009 in African sites as well as in Sri Lanka from an earlier study of *B. dorsalis* were obtained from the molecular biology laboratory at the International Centre of Insect Physiology and Ecology, *icipe* [55,56]. *Wolbachia* infections were initially screened by PCR using the *wsp* primers 81F and 691R [57] and subsequently all positives were screened using the *16S rRNA* primers for *Wolbachia pipientis* [58] and the *Wolbachia* MLST gene primer sets [59]. Reactions were set up in total volumes of 10 µL each, containing 5 × MyTaq reaction buffer (5 mM dNTPs, 15 mM MgCl2, stabilizers and enhancers) (Bioline, London, UK), 2 µM of each primer, 0.25 mM MgCl2 (Thermo Fischer Scientific, Waltham, MA, USA), 0.125 µL MyTaq DNA polymerase (Bioline, London, UK), and 7.5 ng/µL of DNA template. These reactions were set up in a Master cycler Nexus gradient thermo-cycler (Thermo Fischer Scientific, MA, USA). Cycling conditions for the *16S rRNA* primers included an initial denaturation for 2 min at 95 °C, followed by 30 cycles of 1 min at 95 °C, 1 min at 52 °C and 1 min at 72 °C, then a final elongation step of 10 min at 72 °C. For the MLST and *wsp* primers, an initial denaturation for 2 min at 94 °C was used followed by 40 cycles of denaturation of 30 s at 94 °C, 45 s at annealing temperature (55 °C –*wsp*, 54 °C –*hcpA*, *gatB*, *ftsZ*, *coxA* and 59 °C –*fbpA*), 1 min 30 s at 72 °C followed by a final extension step of 10 min at 72 °C. Host mitochondrial DNA was amplified by PCR in similar reaction volumes and cycling conditions as the *wsp* and MLST genes, using the primer LCO1490 and HCO2198 [60] at an annealing temperature of 50.6 °C. PCR products were run through 1% agarose gel electrophoresis and visualized by ethidium bromide staining and UV trans-illumination. Direct sequencing was done for all host COI and *Wolbachia* positive samples. Sequencing was carried out in both directions (F and R) for *Wolbachia* and host COI. *Wolbachia* sequences and representative host haplotype sequences were submitted to the GenBank. Sequence alignments were performed using Clustal W in Geneious 8.1.9 software (www.geneious.com) [61]. Phylogenetic trees were constructed by the neighbor-joining method with the Tamura-Nei model in Geneious 8.1.9 software. Support for tree topology was assessed by bootstrap resampling. A haplotype map was generated using median-joining network algorithm in the population analysis with reticulate trees (popART) software (http://popart.otago.ac.nz) [62,63]. Allelic profiles for *wsp* and MLST sequences obtained were inferred using the *Wolbachia* MLST database (https://pubmlst.org) [59].

## 3. Results

Using a PCR-based assay to amplify the *Wolbachia wsp* gene, we found that out of the 357 individuals tested, 10 were positive for *Wolbachia* (Table 1). These include 6 samples collected between 2005 and 2009 from African sites and 2 collected in Sri Lanka in 2007. In the samples collected in 2017, only 2 were found positive, which corresponds to an overall *Wolbachia* prevalence of 3.6% in the period between 2005 and 2009 and 1.1% in 2017 for the sampled African populations of *B. dorsalis*. Two sites, Muranga and Nguruman, had *Wolbachia* positives in the 2005 to 2009 sample set but not in the 2017 set, whereas Kitui had no positives in the period between 2005 and 2009 but one positive in 2017.

For a total of 6 samples (H-Ng13, Ki1, H-Mu2, Tzc13, H-Tg6 and H-Sl11), we amplified the *Wolbachia coxA* gene in addition to *wsp,* whereas in fewer samples (Ki1, Ng13, Mu2 and Sl11) other MLST genes were amplified and sequenced (Table 2). A full MLST profile and 16SrDNA was achieved for one sample (Tzc13), which had identical allelic profiles in their *wsp* hypervariable regions and *coxA* locus to strains in the *Wolbachia* pubMLST database, however, none of Tzc13’s other loci were identical to those in previously characterized *Wolbachia* strains (Appendix A). The rest of the positive samples had incomplete profiles and partial similarities to those in the pubMLST database.

To investigate the phylogenetic relationship between the detected *Wolbachia,* we constructed a phylogenetic tree with 41 sequences available in GenBank. We used *wsp* sequences for 5 African samples and 1 Sri Lankan sample for which we were able to amplify at least one additional MLST gene. None of these *Wolbachia* were identical to previously detected *Wolbachia* strains in *B. dorsalis* or other *Bactrocera* species. The *wsp* sequences clustered with other *Wolbachia* strains in super group B (Figure 1). Two samples collected in Kenya in the period between 2005 and 2009 (H-Ng3 and H-Mu2) were found to have identical *wsp* sequences, whereas all other samples had *wsp* gene sequences that were unique within *B. dorsalis*.

Amongst the MLST loci, the *coxA* was amplified in the greatest number of samples. The phylogenetic relationships between strains as inferred by *coxA* gene sequence are similar to those inferred using the *wsp* gene with one notable exception; the *coxA* gene sequence for H-Sl11 clustered with *Wolbachia* group A as opposed to its *wsp* gene sequence, which clusters with group B (Figure 2). We observed that samples H-Ng3 and H-Mu2, which had identical *wsp* sequences, also had identical *coxA* sequences.

Sequences from other common *Wolbachia* strains (wPip, wNo and wRi) are included, while a sequence from *Wolbachia* endosymbiont of *Brugia Malayi* is included as an outgroup. *Wolbachia* super groups are indicated in higher case letters on the right. The detected *Wolbachia* are denoted by W followed by the host denoted as dor (*B. dorsalis*), population (Tg-Togo, Tzc-Tanzania, Ki-Kitui, Sl-Sri Lanka, Ng-Nguruman and Mu-Muranga) and population sample ID.

In addition to *coxA,* the MLST genes *fbpA, hcpA*, *gatB* and *ftsZ* were amplified in a number of *Wolbachia* positive *B. dorsalis* samples. Phylogenetic inferences based on these genes largely supported the inferences based on *wsp*; H-Mu2 and H-Ng13, which were found to have identical *wsp* and *coxA* sequences were also found identical at the *fbpA* locus (Appendix A). Similarly, the *fbpA* sequence for Ki1 indicated, in agreement with both *wsp* and *coxA,* that this strain belongs to super group B. However, there were a few notable exceptions; in samples H-Sl11 and Tzc13, which were B super group *Wolbachia* based on the *wsp* locus, one or more MLST genes clustered with super group A *Wolbachia* (Appendix A).

The samples collected in 2017 clustered into 7 mitochondrial *COI* haplotypes (Hap1-Hap7), whereas samples collected between 2005 and 2009 clustered into five of the aforementioned haplotypes (with the exception of Hap5 and 7) and an additional 12 smaller haplotypes (Figure 3). The *COI* gene sequences of the *Wolbachia*-infected *B. dorsalis* from African populations indicated that all were either Hap1 (H-Ng13, Ki1 and Tzc13) or Hap2 (H-Mu2 and H-Tg6). The infected Sri Lankan sample (H-Sl11) did not cluster into any of the 7 major and 12 minor mitochondrial *COI* haplotypes known from Africa. The two samples that had identical *wsp* and *coxA* gene sequences (H-Ng13 and H-Mu2) were found to have different mitochondrial *COI* haplotypes.

In sites that were sampled in both the 2005 to 2009 period and 2017, the most dominant haplotype in the 2005 to 2009 population was also dominant in the 2017 populations (Figure 4), inferring minimal change in the population structures as inferred by mitochondrial haplotype.

## 4. Discussion

We investigated *Wolbachia* infections in *B. dorsalis* in Africa and one location outside of Africa. *Wolbachia* sequences were detected in 10 samples. Overall, this indicates a low rate of *Wolbachia* infection across the African populations, although infection rates appear to be marginally higher than found in Asia [52,53]. It is notable that a higher prevalence of *Wolbachia* infection was observed in Sri Lanka, which is within the native range for this species. Based on their *wsp* and *coxA* sequence, four distinct variants of *Wolbachia*: WdorTg6, WdorTzc13, WdorKi1 and WdorNg3/WdorMu2 were detected in the African populations of *B. dorsalis*. A fifth variant, WdorSl11, was detected in one of the Sri Lankan samples. All of these variants were distinct from those previously observed in *B. dorsalis* [52,53]; this therefore suggests that this species has a high diversity of low prevalence *Wolbachia*.

The WdorNg3/WdorMu2 variant was detected in two individuals from two different sampling sites, Nguruman and Muranga, which are within geographically-separated agro-ecological zones in Kenya. It was notable that this variant was observed in two different mitochondrial haplotypes (Hap1 and Hap2). *Wolbachia* strains only transmitted from an infected female to her offspring (strict vertical transmission) tend to associate strictly with the same host haplotypes while strains that are occasionally transmitted horizontally (from one individual to another unrelated individual) do not. Altogether, the non-concordance with mitochondrial DNA, suggests that some of the *Wolbachia* strains infecting African populations of *B. dorsalis* may have an appreciable rate of horizontal transmission.

In four out of the ten *wsp* positive samples, we were not able to amplify any of the MLST or the *16S rRNA* genes. This may be indicative of transient infections that could have spilled over from an unknown source, or less likely they could be related to a partial transfer of genes (more common than the transfer of a full *Wolbachia* genome) from the symbiont to the host genome. This phenomenon has been observed in previous studies [49], where *Wolbachia* pseudogenes have been detected in other tephritid fruit flies. The nature of these infections could be examined by monitoring vertical transmission rates and tissue localization patterns. We also cannot rule out the possibility that sample trapping and storage may have caused DNA degradation, which could have led to some false *Wolbachia* negative samples and an inability to amplify all loci for the *Wolbachia* positive samples.

At the majority of the loci investigated, the *Wolbachia* variants detected among the African populations clustered with super group B *Wolbachia*, which differs from the majority of super group A strains as found in *B*. *dorsalis* species in China [53] and Thailand [52]. For sample Tzc13, one MLST locus (*fbpA*) segregated with group A *Wolbachia*, suggesting a possible recombination event between an A and B group strains, a phenomenon that has been reported in previous studies [64,65]. For another sample, H-Sl11, we observed that the *wsp* gene segregated with super group B *Wolbachia* in contrast to the *coxA* and *hcpA* loci which segregated as super group A *Wolbachia*. However, sequencing results of the *16S rRNA*, *gatB* and *ftsZ* loci of H-Sl11 (data not shown) revealed an interfering secondary chromatogram in few segments suggesting that the sample is likely to have been infected by two *Wolbachia* strains (A and B group).

In addition, none of the detected *Wolbachia* had complete identity of allelic profiles to *Wolbachia* strains in the *Wolbachia* MLST database. Full length sequences of each of the MLST and *wsp* marker could not be retrieved for all the samples, and this limited our ability to confirm their designation as novel strain types through allelic profiles.

Some strains of *Wolbachia* are known to cause male killing, and in insect populations with these strains, infected male specimens are generally not observed or observed only very rarely. To avoid bias against the detection of male-killing *Wolbachia*, female specimens are generally screened. *Wolbachia* are also known to reach high densities in insect ovaries and therefore may be easier to detect in females. Male lures are among the most widely used attractants in trapping of tephritid fruit flies. Therefore, the majority of the samples we screened, in particular from the 2005 to 2009 collections, were male. It is possible that if female flies had been screened, a higher *Wolbachia* prevalence rate would have been observed. However, it is noteworthy that the female samples from the 2017 collections did not have a higher rate of *Wolbachia* infection. Similar observations have been reported among *B. dorsalis* populations in Asia where the sexual differences in this host does not influence the detection rate of *Wolbachia* [53]. The *Wolbachia* variants that were found in males (WdorTg6, WdorTzc13, WdorNg3/WdorMu2 and WdorSl11) are unlikely to have a male-killing phenotype, whereas we cannot rule out the possibility that WdorKi1, which was detected in a female fly, causes male killing.

The population structure of *B. dorsalis* in the African region was observed to be largely unchanged between the 2005 to 2009 and 2017 samples, with two major mtCOI haplotypes (Hap1 and Hap2) dominating in both [56]. This suggests that based on mitochondrial DNA diversity, the current population structure has been largely unchanged since the initial *B. dorsalis* invasion in the Eastern Africa region, and that no significant new reintroductions with new mtCOI haplotypes have occurred in this region since the first invasion. To confirm this, however, it would be necessary to use nuclear DNA markers as mitochondrial DNA introgression could have occurred. The two main haplotypes, Hap1 and Hap2, in addition to Hap3, Hap4 and Hap7 were observed to be fairly distributed across the East and West of Africa. Only two haplotypes: Hap5 and Hap6 were observed in Eastern Africa only while the rest of the smaller mtCOI haplotypes (Hap8 to Hap18) were observed in West Africa, particularly in Nigeria, except for Hap8 that was observed exclusively in Benin. Previous microsatellite genotyping data also revealed differences between the Eastern Africa populations with that of Nigeria, which were more closely allied to the diverse Sri Lankan populations of *B. dorsalis* [56]. It has been suggested that contemporary gene flow may have contributed to this diversity in West Africa [56].

## 5. Conclusions

We detected four different *Wolbachia* variants in African populations of the oriental fruit fly. Analysis of the host mtCOI haplotypes did not reveal a link between a particular *Wolbachia* variant and host haplotypes. Only the two dominant haplotypes were found to be infected with *Wolbachia* in Africa. These *Wolbachia* should be investigated for their capacity to manipulate host reproduction and to confer hosts with differential susceptibility to parasitoids and pathogens. A comprehensive understanding of the role of *Wolbachia* in this species could improve the effectiveness of integrated control strategies and eventually play a role in the sustainable pest management of *B. dorsalis*.

## Figures and Tables

**Figure 1 insects-10-00155-f001:**
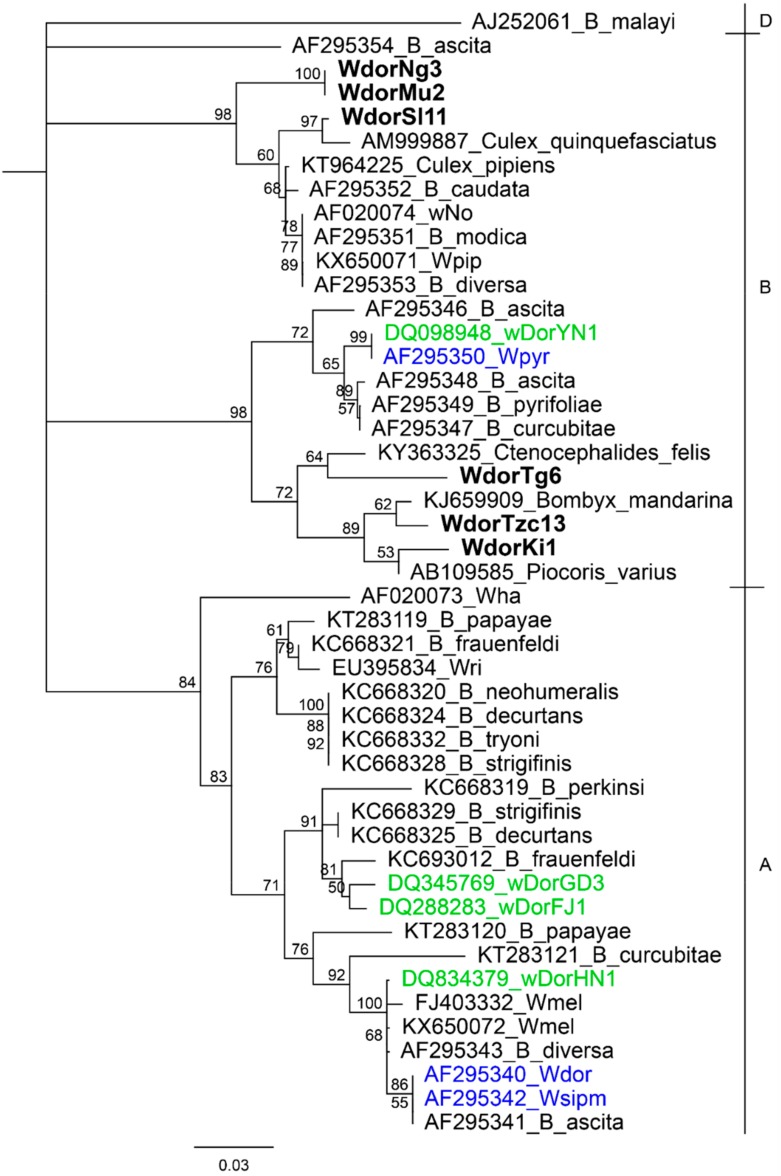
Neighbour joining tree based on *Wolbachia* surface protein (*wsp*) gene sequences of *Wolbachia* detected from *B. dorsalis* in this study (in bold), from *B. dorsalis* in China (labelled in blue), from *B. dorsalis* in Thailand (labelled in green) and from other *Bactrocera* species. Sequences from closest homology matches to *Wolbachia* detected in this study are also included. Other common *Wolbachia* strains (wMel, wRi, wHa, wNo and wPip) are also included. *Wsp* sequence from *Wolbachia* endosymbiont of *Brugia malayi* is included as an outgroup. Sequences are labelled with genbank accession numbers followed by strain name or host organism, or strain name for sequences from this study. *Wolbachia* super groups are indicated in higher case letters on the right. The detected *Wolbachia* are denoted by W followed by the host denoted as dor (*B. dorsalis*), population (Tg-Togo, Tzc-Tanzania, Ki-Kitui, Sl-Sri Lanka, Ng-Nguruman and Mu-Muranga) and population sample ID. Bootstrap values are indicated above branches. Branches with bootstrap support lower than 50% are collapsed.

**Figure 2 insects-10-00155-f002:**
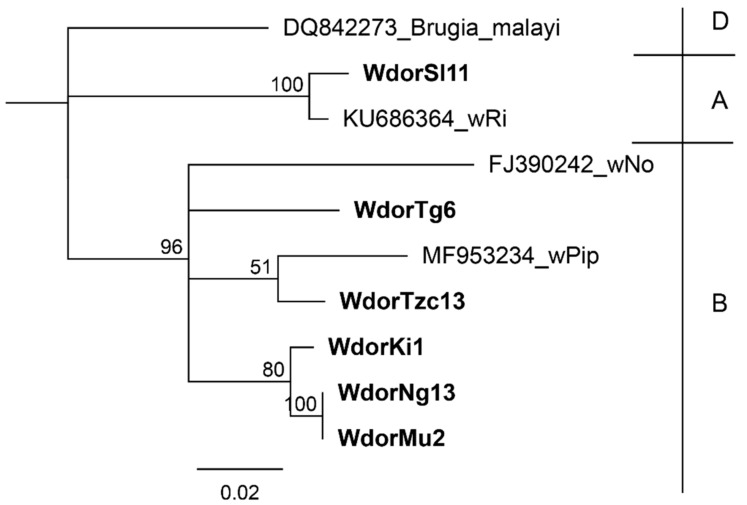
Neighbour joining tree based on the cytochrome c oxidase subunit I (*coxA*) gene sequences from this study. The detected *Wolbachia* are denoted by W followed by the host denoted as dor (*B. dorsalis*), population (Tg-Togo, Tzc-Tanzania, Ki-Kitui, Sl-Sri Lanka, Ng-Nguruman and Mu-Muranga) and population sample ID. Bootstrap values are indicated above branches. Branches with bootstrap support lower than 50% are collapsed.

**Figure 3 insects-10-00155-f003:**
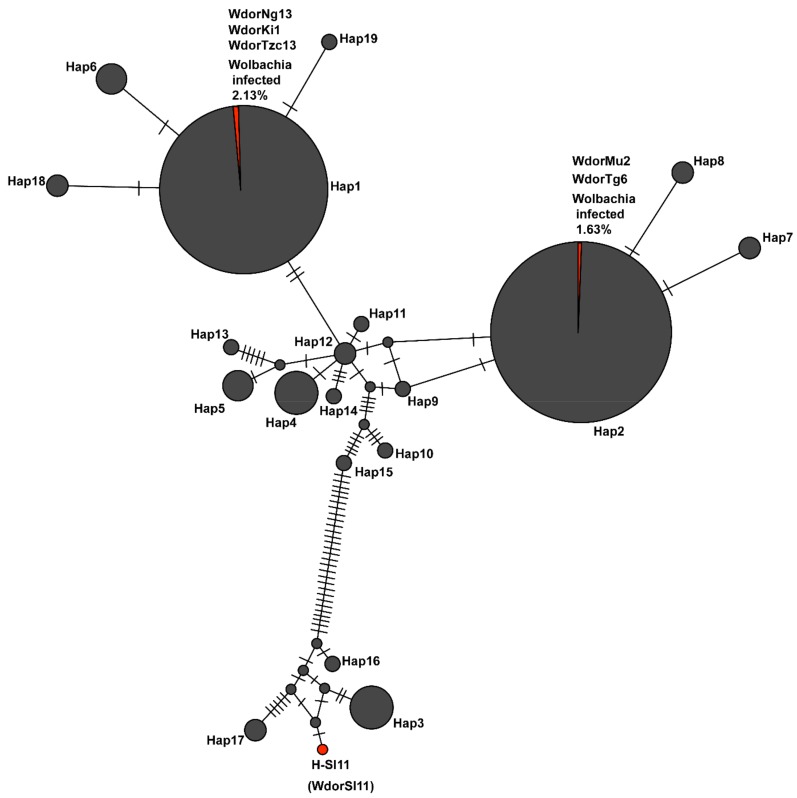
Haplotype map of *Bactrocera dorsalis* mtCOI sequences from African populations. Node size is proportional to number of samples while mutations are represented as hatchmarks. Proportions of *Wolbachia* infected samples (H-Ng13, Ki1 and Tzc13 in Haplotype1 and H-Mu2 and H-Tg6 in Haplotype 2) in their respective haplotypes are shaded in red. Sequences of represented haplotypes are accessible at Genbank using the accessions MK314052-MK31452 for Haplotypes: 3, 2, 4, 6, 7, 5 and 1 respectively, JQ692656, JQ692727, JQ692777, JQ692863, JQ692731, JQ692684, JQ692812, JQ692698, JQ692723, JQ692816, JQ692816, and JQ692691 for haplotypes 8-19 respectively. The infected Sri Lanka sample (labelled H-Sl11, Genbank accession: JQ692764) is not numbered to distinguish it from haplotypes detected in Africa.

**Figure 4 insects-10-00155-f004:**
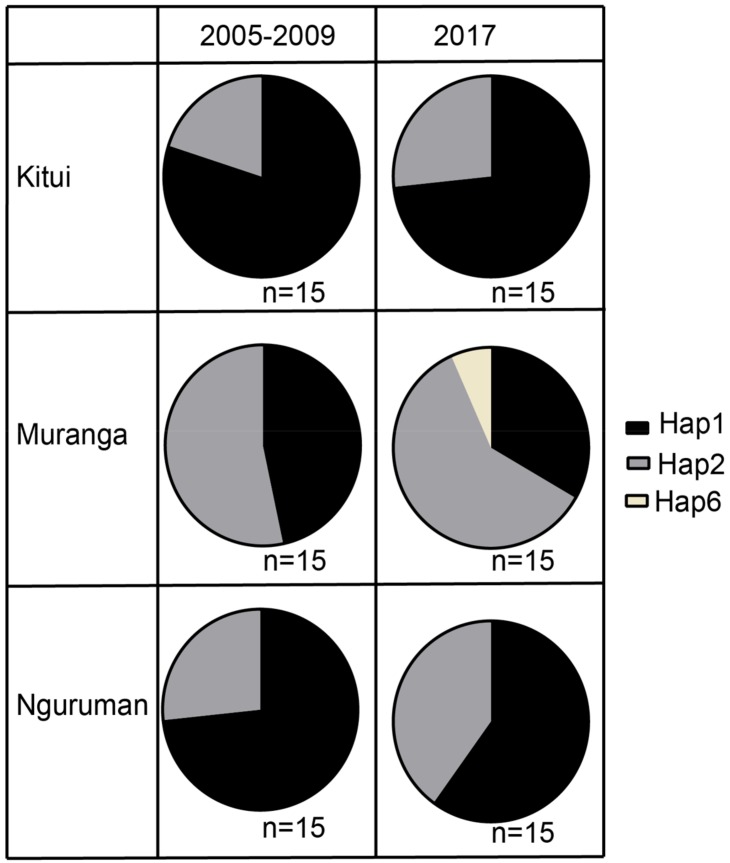
Pie charts depicting haplotype compositions in sites sampled during the period between 2005 to 2009 (left column) and in 2017 (right column).

**Table 1 insects-10-00155-t001:** *B. dorsalis* populations screened for *Wolbachia* using *wsp*. Samples collected between 2005 and 2009 are preceded with “H” to distinguish them from those collected in 2017.

Locality	Specimen Codes	Collection Year/	Collection Sex/(n)	*wsp*+
Nguruman, Kenya	H-Ng	2008	m (15)	1
Nguruman, Kenya	Ng	2017	f (15)	0
Kitui, Kenya	H-Ki	2005	m (15)	0
Kitui, Kenya	Ki	2017	f (15)	1
Muranga, Kenya	H-Mu	2005	m (15)	1
Muranga, Kenya	Mu	2017	f (15)	0
Embu, Kenya	Em	2017	f (15)	0
Dar es Salaam, Tanzania	H-Tz	2009	m (15)	0
Mwanza, Tanzania	Tz-ab	2017	m (30)	0
Morogoro, Tanzania	Tz-c	2017	m (15)	1
Kawanda, Uganda	H-Ug	2007	m (15)	0
Bunamwaya, Uganda	Ug-b	2017	m (30)	0
Khartoum, Sudan	H-Su	2007	m (15)	1
Kassala, Sudan	Su-a	2017	m (15)	0
Gezira, Sudan	Su-b	2017	m (15)	0
Singa, Sudan	Su-c	2017	m (15)	0
Zaria, Nigeria	H-Zr	2005	m (15)	0
Monts Kouffe, Benin	H-Be	2009	m (15)	1
Lome, Togo	H-Tg	2009	m (15)	1
UBG, Ghana	H-Gh	2009	m (15)	1
Ibadan, Nigeria	H-Ib	2009	m (15)	0
Ranbukpitiya, Sri Lanka	H-Sl	2007	m (15)	2

**Table 2 insects-10-00155-t002:** Amplification of 16S rRNA and MLST genes for *wsp* + *B. dorsalis* samples. Numbers represent the number of samples for which there was successful gene amplification.

Samples			*wsp* +	*16S* +	*coxA* +	*fbpA* +	*gatB* +	*hcpA* +	*ftsZ* +
H-Ng13			1	-	1	1	-	-	-
Ki1			1	-	1	-	1	-	-
H-Mu2			1	-	1	1	-	-	-
Tzc13			1	1	1	1	1	1	1
H-Su6			1	-	-	-	-	-	-
H-Be3			1	-	-	-	-	-	-
H-Gh4			1	-	-	-	-	-	-
H-Tg6			1	-	1	-	-	-	-
H-Sl6			1	-	-	-	-	-	-
H-Sl11			1	-	1	-	-	-	-

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
