# Peer review of "Unexpected Diversity of Wolbachia Associated with Bactrocera dorsalis (Diptera: Tephritidae) in Africa"

_insects, 2019, doi:10.3390/insects10060155_

Round 1

Reviewer 1 Report

Overall this is a well written manuscript that provides some interesting results on natural strains of Wolbachia detected in Bactrocera dorsalis.  Although the work presented is well thought out and discussed appropriately, making assumptions based on just wsp and coxA genes is problematic given there are set protocols available for Wolbachia strain characterisation that includes additional genes. The authors would need to do further sequencing of Wolbachia genes to be able to further characterise the phylogeny of these strains and additional gene targets could influence the prevalence rates. 

Further comments are given below: 

Introduction

The introduction is well referenced apart from one section: Lines 54-55. The sentence ‘The use of Wolbachia to reduce the capacity of Aedes mosquitoes to transmit arboviruses has been extensively researched and even implemented’ is missing a significant number of important references and instead includes recent studies that are not appropriate.  More original references needed here on the generation and initial work done with Wolbachia-infected Aedes lines.  

Material and methods

Lines 91-93. Can the authors explain why these two genes were targeted?  The standard procedure for Wolbachia strain characterization is available at https://pubmlst.org/wolbachia/info/protocols.shtml  and this includes wsp primers that are more degenerate and four additional MLST genes in addition to coxA.  Also, 16S is the most common gene target for Wolbachia detection so can the authors explain the rationale for not using this? 

Results

Using the protocols outlined at https://pubmlst.org/wolbachia/wsp/info/protocol.shtml would likely result in different results given these are degenerate wsp primers designed to pick up a larger variety of Wolbachia strains.  It’s also been shown that wsp is not amplified in some species but 16S is more conserved among Wolbachia strains so it normally used for broader detection of strains. I suggest both 16S and degenerate wsp screening is carried out so the authors can also obtain each unique wsp nucleotide allele and each unique HVR amino acid sequence through submission.  If there was a complete profile obtained from all five MLST genes then this would less important but as coxA was only done then wsp typing needs further work to allow conclusions on strain diversity to be made.

Lines 119-128: your wsp phylogenetic tree would indicate novel strains of Wolbachia so further characterisation is required for confirmation.  

Lines 143-145. ‘We amplified and sequenced the Wolbachia coxA gene from the same 5 African and 1 Sri Lankan samples that had their wsp gene sequenced (Figure 2). The phylogenetic relationships between strains as inferred by coxA gene sequence was not fully identical to that inferred by the wsp gene.’  Why was the CoxA gene selected and not the other four MLST genes?  Also, the reason all five MLST genes are needed is to make sure conclusions on phylogeny are not based on a single gene so this needs to be done.  If novel alleles for the MLST genes are present this will be seen with submission to the pubMLST allowing conclusions to be made on the novelty of strains. 

Lines 158-164:Although this is interesting it’s not expected to see any correlation and you would need more of the MLST genes to make any conclusions.  Therefore, this analysis would need to be done after obtaining the concatenated MLST sequences for these conclusions to be fully made.  

Discussion:

Lines 184-187: was this other screening work carried out with wsp 81/691 too?  If it was 16S or other genes, it might not be appropriate to make this comparison on prevalence rates. 

Lines 221-230. The argument that screening females instead of males would allow greater detection would indicate that these strains are low density.  Could you not use qPCR targeting either wsp, 16S or ftsZ to show this is the case? 

Line 239:I don’t think you can say ‘We detected four different strains of Wolbachia in African populations of the oriental fruit fly’ given your lack of strain typing.  More MLST genes would be needed to make this conclusion. 

Author Response

Reviewer 1

The introduction is well referenced apart from one section: Lines 54-55. The sentence ‘The use of Wolbachia to reduce the capacity of Aedes mosquitoes to transmit arboviruses has been extensively researched and even implemented’ is missing a significant number of important references and instead includes recent studies that are not appropriate.  More original references needed here on the generation and initial work done with Wolbachia-infected Aedes lines.  

Indeed, we had cited some recent reviews/studies instead of the more key original work. We have amended this to include the key original papers describing the generation of Dengue-blocking Wolbachia /Aedes mosquitoes and a recent paper on the implementation/deployment of this strategy.

Material and methods

Lines 91-93. Can the authors explain why these two genes were targeted?  The standard procedure for Wolbachia strain characterization is available at https://pubmlst.org/wolbachia/info/protocols.shtml and this includes wsp primers that are more degenerate and four additional MLST genes in addition to coxA.  Also, 16S is the most common gene target for Wolbachia detection so can the authors explain the rationale for not using this?

Using the protocols outlined at https://pubmlst.org/wolbachia/wsp/info/protocol.shtml would likely result in different results given these are degenerate wsp primers designed to pick up a larger variety of Wolbachia strains.  It’s also been shown that wsp is not amplified in some species but 16S is more conserved among Wolbachia strains so it normally used for broader detection of strains. I suggest both 16S and degenerate wsp screening is carried out so the authors can also obtain each unique wsp nucleotide allele and each unique HVR amino acid sequence through submission.  If there was a complete profile obtained from all five MLST genes then this would less important but as coxA was only done then wsp typing needs further work to allow conclusions on strain diversity to be made.

We had initially started by screening B. dorsalissamples with the common 16S primers ‘wspec’ from Werren and Windsor (2000). We concluded that these primers were resulting in false positives as a very significant amount of the samples that had amplification, when sequenced turned out to be not actually Wolbachia. In light of this we decided to opt for the Wsp primers that are known to be effective at detecting the more common A and B group Wolbachia.

We attempted to generate full MLST profiles for all theWolbachiapositive samples, however we were only able to get amplification and good quality sequence for coxA and Wsp. This could be due to some degradation in the quality of the DNA for some specimens which were collected 10+ years ago. We note that a number of other studies have experienced challenges in obtaining full MLST complements. (e.g. https://www.ncbi.nlm.nih.gov/pmc/articles/PMC3236762/) and that there have been some questions raised as to utility of obtaining sequence for all MLST genes (https://academic.oup.com/femsec/article/94/1/fix163/4654844). We note that full MLST profiles are still considered the standard in the field to consider a Wolbachiastrain to be ‘novel’ and we have therefore not referred to the strains described in this paper as ‘novel’.

Lines 119-128: your wsp phylogenetic tree would indicate novel strains of Wolbachia so further characterization is required for confirmation.  

See above.

Lines 143-145. ‘We amplified and sequenced the Wolbachia coxA gene from the same 5 African and 1 Sri Lankan samples that had their wsp gene sequenced (Figure 2). The phylogenetic relationships between strains as inferred by coxA gene sequence was not fully identical to that inferred by the wsp gene.’  Why was the CoxA gene selected and not the other four MLST genes?  Also, the reason all five MLST genes are needed is to make sure conclusions on phylogeny are not based on a single gene so this needs to be done.  If novel alleles for the MLST genes are present this will be seen with submission to the pubMLST allowing conclusions to be made on the novelty of strains. 

CoxA was selected as were able to amplify and sequence this gene from most samples. Amplification and sequencing was also attempted with:

ftsZ

hcpA

fbpA

However, we only got amplification with a few of the samples which rendered these genes of little utility for comparisons within our sample set.

Lines 158-164: Although this is interesting it’s not expected to see any correlation and you would need more of the MLST genes to make any conclusions.  Therefore, this analysis would need to be done after obtaining the concatenated MLST sequences for these conclusions to be fully made.

We agree that MLST data would be useful for making further conclusions, due to the issues raised above we do not believe it will be feasible to get full MLST profiles from these samples.

Line 239: I don’t think you can say ‘We detected four different strains of Wolbachia in African populations of the oriental fruit fly’ given your lack of strain typing. More MLST genes would be needed to make this conclusion. 

We agree that full MLST and hvr profiling would be required for strain identification. We have amended the manuscript such that it does not imply we confirmed the specific identity of the strains in question. We refer to variants as opposed to ‘strains’.

Discussion:

Lines 184-187: was this other screening work carried out with wsp 81/691 too?  If it was 16S or other genes, it might not be appropriate to make this comparison on prevalence rates.

Yes, both previous studies also used the same primer pair.

Lines 221-230. The argument that screening females instead of males would allow greater detection would indicate that these strains are low density.  Could you not use qPCR targeting either wsp, 16S or ftsZ to show this is the case? 

We believe that these infections are likely to be low density infection (in part due to difficulties in obtaining full MLST profiles). Given that the specimens were extracted by different methods and some had been stored for many years we doubted that we could generate meaningful comparisons of density. 

Reviewer 2 Report

“Unexpected diversity of Wolbachia associated with Bactrocera dorsalis (Diptera: Tephritidae) in Africa – Detection and strain identity” by Gichuhi et al., submitted to Insects.

This paper describes the Wolbachia infection and its association with mtDNA haplotype variations in African populations of Bactrocera dorsalis, an important pest of fruits. I consider that this paper provides important data in respect of insect science and pest management, and hence, undoubtedly worth publication in Insects.

I have several concerns regarding the neutrality of presentation, which should be responded by the authors.

Is this the first report of the mtDNA haplotype variation of B. dorsalis on African Continent? If so, explicitly say so in the manuscript. If not, please cite the previous reports and include those data in analyses (if possible). Apparently, there are previous reports on mtDNA variations of B. dorsalis in China, such as Environ. Entomol. 34:977–983 (2005) and BMC Evol. Biol. 12:130 (2012) (and some more?). Why not cite these data and discuss the invasion of this species (as well as Wolbachia infection) from evolutionary viewpoint? Because polymorphic nuclear DNAs were not sequenced, possible existence of cryptic species cannot be validated. The authors should discuss this possibility (unless otherwise denied by other data) and also explicitly mention this limitation (i.e., lack of nuclear DNA data).

Minor comments

-       Title: “Detection and strain identity”. Is it really necessary?

-       In this study, 357 individuals were tested for Wolbachia and 10 were positive (Line 106-107). I suggest including this information in Abstract.

-       Figure 2: The figure and the legend are not matched. For example, shouldn’t it be WdorSl11 instead of Sl11?

-       Figure 2: MF953234_Wpip -> MF953234_wPip

-       Figure 2: Looks like branches supported by bootstrap values of less than 50% were collapsed, nothing is mentioned about this.

-       Figure 2: If Brugia malayi is the outgroup, the horizontal bar and 100 at the left are meaningless (unnecessary).

-       Figure 3: Why only some of the haplotypes are numbered?

-       Figure 3: Nine haplotypes at the bottom are distantly related to the others. Can you say something about gene flow? Is it possible that these insects are cryptic species? Nuclear sequence data may clarify this point.

-       Figure 3: Are there any geographical patterns in mtDNA haplotypes? Readers can easily see this point by connecting the data with Table 1.

-       Reference 1: Volume number is missing.

-       Reference 15: Sc -> Science

-       Reference 24: Journal name and volume number are missing.

-       Reference 31: Page numbers are missing.

-       Reference 56: Neill, S.L.O. -> O’Neill, S.L.; Curr micro -> Curr. Microbiol.

-       Reference 58: Journal name is missing.

Author Response

Reviewer 2

Is this the first report of the mtDNA haplotype variation of B. dorsalis on African Continent? If so, explicitly say so in the manuscript. If not, please cite the previous reports and include those data in analyses (if possible). Apparently, there are previous reports on mtDNA variations of B. dorsalis in China, such as Environ. Entomol. 34:977–983 (2005) and BMC Evol. Biol. 12:130 (2012) (and some more?). Why not cite these data and discuss the invasion of this species (as well as Wolbachia infection) from evolutionary viewpoint? Because polymorphic nuclear DNAs were not sequenced, possible existence of cryptic species cannot be validated. The authors should discuss this possibility (unless otherwise denied by other data) and also explicitly mention this limitation (i.e., lack of nuclear DNA data).

The first report of mtDNA haplotypes in Africa was Khamis et. al. 2012 (using some of the samples we analyzed in this paper). An earlier paper (Khamis 2009) carried out a microsatellite analysis. Together, these are a comprehensive investigation of African diversity of this species, cryptic sub-species and invasion dynamics. We have also mentioned the limitations of our data (lack of nuclear DNA).

Minor comments

-       Title: “Detection and strain identity”. Is it really necessary?

We accepted to remove the phrase from the title

-       In this study, 357 individuals were tested for Wolbachia and 10 were positive (Line 106-107). I suggest including this information in Abstract.

This information has been added to the abstract

-       Figure 2: The figure and the legend are not matched. For example, shouldn’t it be WdorSl11 instead of Sl11?

The corrections have been made in the figure

-       Figure 2: MF953234_Wpip -> MF953234_wPip

The corrections have been made in the figure

-       Figure 2: Looks like branches supported by bootstrap values of less than 50% were collapsed, nothing is mentioned about this.

The corrections have been made in the legend

-       Figure 2: If Brugia malayi is the outgroup, the horizontal bar and 100 at the left are meaningless (unnecessary).

The corrections have been made in the figure

-       Figure 3: Why only some of the haplotypes are numbered?

We numbered only the haplotypes observed among the African samples, we avoided numbering the Sl11 sample to avoid seeming like it was also a haplotype in Africa. We have added this disclaimer to the figure legend.

-       Figure 3: Nine haplotypes at the bottom are distantly related to the others. Can you say something about gene flow? Is it possible that these insects are cryptic species? Nuclear sequence data may clarify this point.

We have included a paragraph in the discussion about this, linking to previous work which provides a much more exhaustive investigation of potential cryptic species.

-       Figure 3: Are there any geographical patterns in mtDNA haplotypes? Readers can easily see this point by connecting the data with Table 1.

We have included in the discussion the haplotype distribution in the continent.

 -       Reference 1: Volume number is missing.

This has been corrected

-       Reference 15: Sc -> Science

This has been corrected

-       Reference 24: Journal name and volume number are missing.

This has been corrected

-       Reference 31: Page numbers are missing.

This has been corrected

-       Reference 56: Neill, S.L.O. -> O’Neill, S.L.; Curr micro -> Curr. Microbiol.

This has been corrected

-       Reference 58: Journal name is missing.

This has been corrected

Reviewer 3 Report

Bactrocera dorsalis from Africa was screened for Wolbachia and strains discovered were compared with those already found in other organisms using phylogenetic methods. Overall, Wolbachia strains collected were diverse and not identical to those those known from Bactrocera or its parasitoids, but they occurred at low frequency and were not strictly associated with mitochondrial haplotypes of B. dorsalis. The study is well-written and presented clearly. 

Some comments to be addressed follow here:

What was the reason for choosing wsp and coxA to screen rather than using the full MLST system of Baldo et al. (2006)? The reason for your choice should be included in the Methods.

Some samples in Table 1 tested positive using primers for wsp, but not for coxA. What would be an explanation for this and why were the wsp amplicons not sequenced for these samples?

Line 198. It would be good to define horizontal transmission and vertical transmission of Wolbachia.

Line 234 onwards. Would it be possible that multiple introductions have come into Africa, but from the same source and same mitochondrial haplotype? How do the mitochondrial haplotypes identified in this study compare with those from possible source populations in Asia or elsewhere?

Author Response

Reviewer 3

What was the reason for choosing wsp and coxA to screen rather than using the full MLST system of Baldo et al. (2006)? The reason for your choice should be included in the Methods.

We attempted to generate full MLST profiles for all the Wolbachia positive samples, however we were only able to get amplification and good quality sequence for coxA and Wsp. This could be due to some degradation in the quality of the DNA for some specimens which were collected 10+ years ago. We have mentioned this in the methods.

Some samples in Table 1 tested positive using primers for wsp, but not for coxA. What would be an explanation for this and why were the wsp amplicons not sequenced for these samples?

There may be a number of possible explanations for this. Most likely the sensitivity / detection threshold differs between these primer pairs and DNA degradation may result in some primers not detecting Wolbachia. Another possible explanation is that fragments (incomplete genome) of Wolbachiahave been inserted in the host genome (horizontal gene transfer, HGT). We decided to focus efforts on the Wolbachiasfor which we could detect more than one gene given that these are more likely to be ‘real’ infections and not remnants of past HGT events. 

Line 198. It would be good to define horizontal transmission and vertical transmission of Wolbachia.

This has been included in the discussion.

Line 234 onwards. Would it be possible that multiple introductions have come into Africa, but from the same source and same mitochondrial haplotype? How do the mitochondrial haplotypes identified in this study compare with those from possible source populations in Asia or elsewhere?

These are important questions that have been largely addressed by previously published studies. The first report of mtDNA haplotypes in Africa was Khamis et. al. 2012 (using some of the samples we analyzed in this paper). An earlier paper (Khamis 2009) carried out a microsatellite analysis. Together, these are a comprehensive investigation of African diversity of this species, cryptic sub-species and invasion dynamics.

Round 2

Reviewer 1 Report

Reviewer original comment:

The introduction is well referenced apart from one section: Lines 54-55. The sentence ‘The use of Wolbachia to reduce the capacity of Aedes mosquitoes to transmit arboviruses has been extensively researched and even implemented’ is missing a significant number of important references and instead includes recent studies that are not appropriate.  More original references needed here on the generation and initial work done with Wolbachia-infected Aedes lines.  

authors response:  

Indeed, we had cited some recent reviews/studies instead of the more key original work. We have amended this to include the key original papers describing the generation of Dengue-blocking Wolbachia /Aedes mosquitoes and a recent paper on the implementation/deployment of this strategy.

Reviewer response: 

You had five references for this point in the original ms and you now have four "The use of Wolbachia to reduce the capacity of Aedes mosquitoes to transmit arboviruses has been extensively researched and even implemented [16–19]" and there are more appropriate references that could have been included.

Reviewer original comment:

Lines 91-93. Can the authors explain why these two genes were targeted?  The standard procedure for Wolbachia strain characterization is available at https://pubmlst.org/wolbachia/info/protocols.shtml and this includes wsp primers that are more degenerate and four additional MLST genes in addition to coxA.  Also, 16S is the most common gene target for Wolbachia detection so can the authors explain the rationale for not using this?

Using the protocols outlined at https://pubmlst.org/wolbachia/wsp/info/protocol.shtml would likely result in different results given these are degenerate wsp primers designed to pick up a larger variety of Wolbachia strains.  It’s also been shown that wsp is not amplified in some species but 16S is more conserved among Wolbachia strains so it normally used for broader detection of strains. I suggest both 16S and degenerate wsp screening is carried out so the authors can also obtain each unique wsp nucleotide allele and each unique HVR amino acid sequence through submission.  If there was a complete profile obtained from all five MLST genes then this would less important but as coxA was only done then wsp typing needs further work to allow conclusions on strain diversity to be made.

 authors response

We had initially started by screening B. dorsalissamples with the common 16S primers ‘wspec’ from Werren and Windsor (2000). We concluded that these primers were resulting in false positives as a very significant amount of the samples that had amplification, when sequenced turned out to be not actually Wolbachia. In light of this we decided to opt for the Wsp primers that are known to be effective at detecting the more common A and B group Wolbachia.

We attempted to generate full MLST profiles for all theWolbachiapositive samples, however we were only able to get amplification and good quality sequence for coxA and Wsp. This could be due to some degradation in the quality of the DNA for some specimens which were collected 10+ years ago. We note that a number of other studies have experienced challenges in obtaining full MLST complements. (e.g. https://www.ncbi.nlm.nih.gov/pmc/articles/PMC3236762/) and that there have been some questions raised as to utility of obtaining sequence for all MLST genes (https://academic.oup.com/femsec/article/94/1/fix163/4654844). We note that full MLST profiles are still considered the standard in the field to consider a Wolbachia strain to be ‘novel’ and we have therefore not referred to the strains described in this paper as ‘novel’.

Reviewer response: 

Firstly, in my experience it is extremely rare not to be able to successful amplify Wolbachia strains using the 16S primers and there are many examples where even very low density strains still amplify this conserved gene target.  These results would need to be discussed further in the manuscript given these are surprising results. 

Secondly, only amplifying coxA and wsp is again something that doesn't make sense to me given different Wolbachia strains will have genetic variability so it seems unlikely that these two genes were the only ones that worked for all 'four strains'.  It seems more logical that some of the samples amplified other MLST genes so the authors should provide the amplification profiles of individual samples and also provide the sequences of any successful sequences (regardless of whether all 'four' strains had successfully sequencing for that gene).  

Thirdly, the authors have not considered further wsp typing to obtain unique wsp nucleotide allele and unique HVR amino acid sequences so this limits conclusions using the wsp sequences presented in this manuscript.  

Although I agree that full MLST is not always possible but 1/5 genes and no 16S sequences (despite wsp) is unusual and if DNA degradation has occurred then does this not question some of the conclusions you can make on the phylogeny of these strains? The manuscript PMC3236762 actually argues that ' whole genome typing approaches should be used for Wolbachia typing in the future' and surely providing more than one MLST gene sequence provides more phylogenetic information than just CoxA?

Reviewer original comment:

Lines 143-145. ‘We amplified and sequenced the Wolbachia coxA gene from the same 5 African and 1 Sri Lankan samples that had their wsp gene sequenced (Figure 2). The phylogenetic relationships between strains as inferred by coxA gene sequence was not fully identical to that inferred by the wsp gene.’  Why was the CoxA gene selected and not the other four MLST genes?  Also, the reason all five MLST genes are needed is to make sure conclusions on phylogeny are not based on a single gene so this needs to be done.  If novel alleles for the MLST genes are present this will be seen with submission to the pubMLST allowing conclusions to be made on the novelty of strains. 

authors response: 

CoxA was selected as were able to amplify and sequence this gene from most samples. Amplification and sequencing was also attempted with:

ftsZ

hcpA

fbpA

However, we only got amplification with a few of the samples which rendered these genes of little utility for comparisons within our sample set.

Reviewer response:

I appreciate that maybe amplification was not successful for all samples but for those that worked these new sequences could be used to put your strains in context with fully characterised strains (and not just for comparison to just the strains you are presenting in the manuscript).  

Reviewer original comment:

Lines 221-230. The argument that screening females instead of males would allow greater detection would indicate that these strains are low density.  Could you not use qPCR targeting either wsp, 16S or ftsZ to show this is the case? 

authors response: 

We believe that these infections are likely to be low density infection (in part due to difficulties in obtaining full MLST profiles). Given that the specimens were extracted by different methods and some had been stored for many years we doubted that we could generate meaningful comparisons of density. 

Reviewer response:

Firstly, there are wsp qPCR protocols that would allow you to confirm these low level infections by using methods of normalisation to account for extraction efficiency.  Given you did get amplification of the wsp gene, you could answer this question easily in my opinion. 

Secondly, if they have been extracted using different methods that influence their amplification success would it not question whether phylogenetic relationships between strains can be accurately presented?

Author Response

Reviewer original comment:

The introduction is well referenced apart from one section: Lines 54-55. The sentence ‘The use of Wolbachiato reduce the capacity of Aedes mosquitoes to transmit arboviruses has been extensively researched and even implemented’ is missing a significant number of important references and instead includes recent studies that are not appropriate.  More original references needed here on the generation and initial work done with Wolbachia-infected Aedes lines.  

authors response:  

Indeed, we had cited some recent reviews/studies instead of the more key original work. We have amended this to include the key original papers describing the generation of Dengue-blocking Wolbachia/Aedes mosquitoes and a recent paper on the implementation/deployment of this strategy.

Reviewer response: 

You had five references for this point in the original ms and you now have four "The use of Wolbachia to reduce the capacity of Aedes mosquitoes to transmit arboviruses has been extensively researched and even implemented [16–19]" and there are more appropriate references that could have been included.

This issue was also raised by the editor, who felt that there wasn’t all that much to be gained by referencing the work on Wolbachiain disease vector insects. We have now removed this section in order to shorten the introduction keep its content more relevant to Tephritid fruit flies.

Reviewer original comment:

Lines 91-93. Can the authors explain why these two genes were targeted?  The standard procedure for Wolbachiastrain characterization is available at https://pubmlst.org/Wolbachia/info/protocols.shtml and this includes wsp primers that are more degenerate and four additional MLST genes in addition to coxA.  Also, 16S is the most common gene target for Wolbachiadetection so can the authors explain the rationale for not using this?

Using the protocols outlined at https://pubmlst.org/Wolbachia/wsp/info/protocol.shtml would likely result in different results given these are degenerate wsp primers designed to pick up a larger variety of Wolbachiastrains.  It’s also been shown that wsp is not amplified in some species but 16S is more conserved among Wolbachiastrains so it normally used for broader detection of strains. I suggest both 16S and degenerate wsp screening is carried out so the authors can also obtain each unique wsp nucleotide allele and each unique HVR amino acid sequence through submission.  If there was a complete profile obtained from all five MLST genes then this would less important but as coxA was only done then wsp typing needs further work to allow conclusions on strain diversity to be made.

 authors response

We had initially started by screening B. dorsalis samples with the common 16S primers ‘wspec’ from Werren and Windsor (2000). We concluded that these primers were resulting in false positives as a very significant amount of the samples that had amplification, when sequenced turned out to be not actually Wolbachia. In light of this we decided to opt for the Wsp primers that are known to be effective at detecting the more common A and B group Wolbachia.

We attempted to generate full MLST profiles for all the Wolbachia positive samples, however we were only able to get amplification and good quality sequence for coxA and Wsp. This could be due to some degradation in the quality of the DNA for some specimens which were collected 10+ years ago. We note that a number of other studies have experienced challenges in obtaining full MLST complements. (e.g. https://www.ncbi.nlm.nih.gov/pmc/articles/PMC3236762/) and that there have been some questions raised as to utility of obtaining sequence for all MLST genes (https://academic.oup.com/femsec/article/94/1/fix163/4654844). We note that full MLST profiles are still considered the standard in the field to consider a Wolbachia strain to be ‘novel’ and we have therefore not referred to the strains described in this paper as ‘novel’.

Reviewer response: 

Firstly, in my experience it is extremely rare not to be able to successful amplify Wolbachiastrains using the 16S primers and there are many examples where even very low density strains still amplify this conserved gene target.  These results would need to be discussed further in the manuscript given these are surprising results. 

Secondly, only amplifying coxA and wsp is again something that doesn't make sense to me given different Wolbachiastrains will have genetic variability so it seems unlikely that these two genes were the only ones that worked for all 'four strains'.  It seems more logical that some of the samples amplified other MLST genes so the authors should provide the amplification profiles of individual samples and also provide the sequences of any successful sequences (regardless of whether all 'four' strains had successfully sequencing for that gene).  

Thirdly, the authors have not considered further wsp typing to obtain unique wsp nucleotide allele and unique HVR amino acid sequences so this limits conclusions using the wsp sequences presented in this manuscript.  

Although I agree that full MLST is not always possible but 1/5 genes and no 16S sequences (despite wsp) is unusual and if DNA degradation has occurred then does this not question some of the conclusions you can make on the phylogeny of these strains? The manuscript PMC3236762 actually argues that ' whole genome typing approaches should be used for Wolbachia typing in the future' and surely providing more than one MLST gene sequence provides more phylogenetic information than just CoxA?

We respect the reviewers experience in this matter, however we believe that it isn’t that uncommon to have differing levels of success in the amplification of different MLST genes. In some low-density Wolbachiainfections researchers use nested-PCR, however, we feel that this is much more prone to contamination and have therefore tried to avoid it. 

We have now re-screened all the sample set with full MLST gene primers as well as the WSP primers. We have managed to obtain a full MLST profile for one sequence and also type their alleles as well as their wsp HVR regions. Some alleles did not match strains in the database, we have uploaded these to pubmed and indicated that we do have some limitations with respect to typing the alleles of all obtained sequences.

Reviewer original comment:

Lines 143-145. ‘We amplified and sequenced the WolbachiacoxA gene from the same 5 African and 1 Sri Lankan samples that had their wsp gene sequenced (Figure 2). The phylogenetic relationships between strains as inferred by coxA gene sequence was not fully identical to that inferred by the wsp gene.’  Why was the CoxA gene selected and not the other four MLST genes?  Also, the reason all five MLST genes are needed is to make sure conclusions on phylogeny are not based on a single gene so this needs to be done.  If novel alleles for the MLST genes are present this will be seen with submission to the pubMLST allowing conclusions to be made on the novelty of strains. 

authors response: 

CoxA was selected as were able to amplify and sequence this gene from most samples. Amplification and sequencing was also attempted with:

ftsZ

hcpA

fbpA

However, we only got amplification with a few of the samples which rendered these genes of little utility for comparisons within our sample set.

Reviewer response:

I appreciate that maybe amplification was not successful for all samples but for those that worked these new sequences could be used to put your strains in context with fully characterized strains (and not just for comparison to just the strains you are presenting in the manuscript).  

This is an important point that we hope has now been addressed by the inclusion of additional MLST loci. All that could be amplified after repeated attempts are now included.

Reviewer original comment:

Lines 221-230. The argument that screening females instead of males would allow greater detection would indicate that these strains are low density.  Could you not use qPCR targeting either wsp, 16S or ftsZ to show this is the case? 

authors response: 

We believe that these infections are likely to be low density infection (in part due to difficulties in obtaining full MLST profiles). Given that the specimens were extracted by different methods and some had been stored for many years we doubted that we could generate meaningful comparisons of density. 

Reviewer response:

Firstly, there are wsp qPCR protocols that would allow you to confirm these low level infections by using methods of normalisation to account for extraction efficiency.  Given you did get amplification of the wsp gene, you could answer this question easily in my opinion. 

Secondly, if they have been extracted using different methods that influence their amplification success would it not question whether phylogenetic relationships between strains can be accurately presented?

We do not disagree with the reviewer that qPCR could be used with the correct normalization techniques to gain insight into the Wolbachiadensity. If this is deemed of critical importance we will indeed endeavor to carry it out however we think this would be much better suited to a follow-up publication in which we hope to be able to monitor Wolbachiainfected Tephritid flies under laboratory conditions (in the F1 of infected females from the field). In contrast to wild caught specimens, lab-reared flies would be of known ages and therefore the density could be meaningfully compared. Since Wolbachiadensity is known to change very significantly over the lifespan of an insect, it would be difficult to make a conclusion regarding the ‘characteristic’ density of these strains from the data that would be generated by the suggested experiment.

Reviewer 2 Report

I checked that the manuscript has been improved. I do not have any more concerns about this manuscript.

Author Response

Many thanks for reviewing the manuscript an deeming that we have resolved all the issues you had raised.

Round 3

Reviewer 1 Report

The authors have now presented some additional MLST data and I appreciate that not all strains will results in a full MLST profile.  It was good to see this data and I believe it's a much stronger manuscript in it's present form. 

Author Response

We thank the reviewer for his critical assessment of the manuscript.